# A Rare Adult Primary Intracranial Sarcoma, *DICER1*-Mutant Identified by Epigenomic Profiling: A Case Report

**DOI:** 10.3390/brainsci13020235

**Published:** 2023-01-30

**Authors:** Alfredo Marinelli, Mariella Cuomo, Raduan Ahmed Franca, Michela Buonaiuto, Davide Costabile, Cristina Pagano, Federica Trio, Liliana Montella, Maria Laura Del Basso De Caro, Roberta Visconti, Lorenzo Chiariotti, Rosa Della Monica

**Affiliations:** 1Operative Unit Neuroncology, University Federico II, 80131 Naples, Italy; 2Istituto di Ricerca e Cura a Carattere Scientifico (IRCCS), Neuromed Istituto Neurologico Mediterraneo (INM), 86077 Isernia, Italy; 3CEINGE, Advanceed Biotechnolgies “Franco Salvatore”, 80145 Naples, Italy; 4Department of Molecular Medicine and Medical Biotechnologies, University of Naples Federico II, 80131 Naples, Italy; 5Department of Advanced Biomedical Sciences, Pathology Section, University of Naples Federico II, 80131 Naples, Italy; 6SEMM-European School of Molecular Medicine, University of Naples, Federico II, 80145 Naples, Italy; 7ASL NA2 NORD, Oncology Operative Unit, “Santa Maria delle Grazie” Hospital, 80027 Pozzuoli, Italy; 8Institute for the Experimental Endocrinology and Oncology “G. Salvatore”, Italian National Council of Research, 80131 Naples, Italy

**Keywords:** primary intracranial sarcoma, *DICER1*-mutant, methylation tumor classifier, case report

## Abstract

Diagnoses of primary malignant mesenchymal brain tumors are a challenge for pathologists. Here, we report the case of a 52-year-old man with a primary brain tumor, histologically diagnosed as a high-grade glioma, not otherwise specified (NOS). The patient underwent two neurosurgeries in several months, followed by radiotherapy and chemotherapy. We re-examined the tumor samples by methylome profiling. Methylome analysis revealed an epi-signature typical of a primary intracranial sarcoma, *DICER1*-mutant, an extremely rare tumor. The diagnosis was confirmed by DNA sequencing that revealed a mutation in *DICER1* exon 25. *DICER1* mutations were not found in the patient’s blood cells, thus excluding an inherited *DICER1* syndrome. The methylome profile of the *DICER1* mutant sarcoma was then compared with that of a high-grade glioma, a morphologically similar tumor type. We found that several relevant regions were differentially methylated. Taken together, we report the morphological, epigenetic, and genetic characterization of the sixth described case of an adult primary intracranial sarcoma, *DICER1*-mutant to-date. Furthermore, this case report underscores the importance of methylome analysis to refine primary brain tumor diagnosis and to avoid misdiagnosis among morphologically similar subtypes.

## 1. Introduction

Primary malignant mesenchymal brain tumors are a diagnostic challenge because they are rare, similar in clinical presentation, and highly heterogeneous, at both the morphological and genomic levels [1]. Recently, a new “omic” approach based on whole-genome DNA methylation (methylome) analysis has allowed precise classification and subclassification of many tumors on the basis of their “epigenetic signatures” [2]. The methylation-based classifier of brain tumors, developed by Capper et al. in 2018 [3], is constantly updated and implemented by the addition of new entities, including sinonasal tumors and sarcomas [4,5].

Here, we report the case of a primary brain tumor, first diagnosed as a high-grade glioma, NOS (not otherwise specified) in our academic institution. The tumor DNA methylation profile revealed, instead, an epi-signature typical of a primary intracranial sarcoma, *DICER1*-mutant. The diagnosis was confirmed by tissue sample sequencing of the *DICER1* gene. To date, in adults, only five other cases of primary intracranial sarcoma, *DICER1*-mutant, have been reported [5,6,7,8] (Table 1). In addition, this case presentation highlights the importance of methylome analysis to refine primary brain tumor diagnosis and to avoid misdiagnosis among morphologically similar subtypes.

## 2. Case Presentation

### 2.1. Clinical and Radiological Description

In June 2020, a 52-year-old man received emergency surgery because of a cerebral hemorrhage in the left temporo-parietal lobe (Figure 1A). Post-operative magnetic resonance imaging (MRI) revealed a pathological mass in the same left temporo-parietal lobe (Figure 1B). Two months later, the patient experienced a new hemorrhagic episode, and he received a second emergency craniotomy for decompression and subtotal mass reduction (Figure 1C). In December 2020, because of his worsening clinical condition, the patient underwent a new reductive subtotal surgery. In the patient’s pathological history, a diagnosis of malignant histiocytosis was present. In fact, in 1987 he presented with axillary lymphadenopathy. An excisional lymph node biopsy revealed the abovementioned diagnosis. At that time, total-body computed tomography (CT) staging was negative for other localizations.

After the second neurosurgery, the patient received 10 cycles of external-beam radiation therapy (EBRT) at 3 Gray, but radiotherapy was interrupted after 9 cycles because of the poor performance status of the patient. After the last neurosurgery, the patient received chemotherapy based on doxorubicin and dacarbazine. The therapy was interrupted after two cycles because the patient died.

The medical history of the patient is depicted in Figure 2.

### 2.2. Histopathology

As shown in Figure 3, hematoxylin-eosin staining of the lesion excised in December 2020 revealed a highly cellular neoplasm. The tumor cells were spindle-shaped to epithelioid, arranged in a patternless solid growth. There was brisk mitotic activity, also with atypical figures, and geographic necrosis with a “pseudo-palizading pattern”. The tumor was highly hemorrhagic, with abundant foamy histiocytes infiltrating the mass. Upon immunohistochemistry, the tumor cells were found to be GFAP (gliofibrillary acid protein) negative, SMA (smooth muscle actin) negative, IDH1 (isocitrate dehydrogenase1) wild-type, and the cellular proliferative index Ki67 was high, near 70%. A diagnosis of high-grade glioma, NOS, was made.

### 2.3. Classification Based on DNA Methylation Profiling

To refine the histopathological characterization, we performed a DNA methylation profile on the same lesion using the Illumina Technologies EPIC array 850 k.

Formalin-fixed paraffin-embedded (FFPE) tumoral sample DNA was extracted using the FFPE DNA Tissue Kit (Qiagen, Hilden, Germany), following the manufacturer’s instructions. The extracted DNA was converted using bisulfite (EZ gold Kit, Zymo Research, Irvine, CA, USA), following the manufacturer’s instructions. The bisulfite-treated DNA was analyzed using the Illumina EPIC ARRAY 850 Bead-Chip (Infinium human methylation EPIC, Illumina, San Diego, CA, USA), as described in [9]. The array covered 850.000 CpG distributed across the genome. The resulting data (in IDAT format) were processed and analyzed using a previously published bioinformatic algorithm available from the German Cancer Research Center (DKFZ, Heidelberg, Germany) [3]. The DKFZ pipeline for methylome analysis calculated a “calibrated score”, which provided information about the methylation classes and the presence and relative amounts of tumor cells. This score ranges between 0 and 1, where 1 indicates a very high confidence of tumor identity and classification. The generated methylation data were compared with the Heidelberg brain tumor classifier to assign a subgroup score for the tumor compared with 91 different brain tumor entities [10]. The methylation class family is predicted if the score is >0.9, but even a score of >0.5 is sufficient to identify predicted methylation subclasses.

Methylome analysis showed a match (scores > 0.9) with the new *DICER1*-mutant class: primary intracranial sarcoma, *DICER1*-mutant [11]. To further confirm the diagnosis, we also performed a methylation profile analysis of the tumor removed in June 2020. In this case, the methylome analysis showed the epi-signature typical of primary intracranial sarcomas, *DICER1*-mutant. In accordance with the frequent CNV alterations described in primary intracranial sarcomas, *DICER1*-mutant [6], and with the observed high Ki67 index (Figure 3), we found, in both cancer lesions, 1q and 9p chromosome deletions and high amplification of the *CCND2* gene encoding for the D2-type cyclin protein, a strong regulator of cell cycle onset and, thus, of proliferation (Figure 4A). *MGMT* promoter methylation analysis by Methylation Specific PCR (MSP) [12] showed that the *MGMT* promoter was not methylated.

### 2.4. Molecular Investigation: DICER1 and p53 Gene Sequencing

To verify the presence of the *DICER1* mutations predicted by methylome analysis, we performed DNA sequencing of the *DICER1* gene utilizing DNA extracted from the tumor removed in December 2020. As shown in Figure 4B, we found a homozygous, hotspot mutation in *DICER1* exon 25, resulting in the aminoacidic substitution Glu1813Ala. The same mutation was reported in other cases [13]. Thus, *DICER1* DNA sequencing confirmed the diagnosis predicted by the methylome profile. The same Glu1813Ala mutation was also found at 50% in the primary intracranial lesion removed in June 2020. In this case, however, we could not definitively determine the zygosity status because of the heterogeneity in tumor cell composition. *DICER1* variants have been described not only in sporadic tumors but also in familial cancers, patients with germline *DICER1* mutations being predisposed to multiple cancer types [14]. Thus, to address this matter, we sequenced the DNA extracted from the peripheral blood of the patient and from the axillary lymph node eradicated 35 years before at the time of the malignant histiocytosis onset. Sequencing analyses of both blood (Figure 4C) and lymph node did not detect any mutation in *DICER1*. In light of these data, we excluded a germinal mutation in DICER1, demonstrating that tumor DICER1 alterations were caused by two somatic mutational events.

In primary intracranial sarcoma, *DICER1*-mutant, *p53* loss-of-function mutations have frequently been described [5]. However, no *p53* mutations were detected in the two tumor samples we analyzed.

### 2.5. Methylome Profiling of Primary Intracranial Sarcoma, DICER1 Mutant versus Mesenchymal Glioblastoma

Primary intracranial sarcomas, *DICER1*-mutant, are morphologically similar to high-grade gliomas, and thus, the histopathological differential diagnosis may be challenging. To conduct an explorative analysis to determine the extent of the methylation differences, the methylome of the primary intracranial sarcoma, *DICER1*-mutant (surgically removed on December 2020), was compared with the methylome profile of a high-grade glioma. Bioinformatic analyses were performed using RnBeads scripts, as described in [15,16]. Only the CpG sites with a difference of at least 25% in DNA methylation between the primary intracranial sarcoma, *DICER1*-mutant, and the high-grade glioma were filtered and retained. As shown in Figure 5, we identified 20.022 CpGs with a lower degree of DNA methylation in the primary intracranial sarcoma, *DICER1*-mutant, while 8.254 CpG sites were found more methylated in the primary intracranial sarcoma, *DICER1*-mutant, compared with the high-grade glioma. We then associated the differentially methylated CpG sites with the respective genes using the R-based library “GenomicRanges” [17]. Only genes with at least five differentially methylated CpG sites were considered. In this way, as shown in Figure 5, 2.262 hypomethylated and 1.190 hypermethylated genes were identified when comparing the primary intracranial sarcoma, *DICER1*-mutant, and the high-grade glioma.

We then performed KEGG analysis using DAVID [18] to identify the enriched pathways for the differentially methylated genes. In particular, 48 demethylated genes were assigned to the *MAPK* signaling pathway, already described as frequently activated in primary intracranial sarcomas, *DICER1*-mutant, and associated with a high tumor proliferative cellular rate and aggressiveness [6]. Because of the rarity of these tumor entities, the data available so far, it is not possible to obtain robust results; however, the above data suggest that the differences between the two entities are well reflected by the methylation signatures.

## 3. Discussion

Primary intracranial sarcomas, *DICER1*-mutant, are rare neoplasms identified and typified through analysis of the whole-genome DNA methylation profile. To date, very few cases have been described, and these were predominantly primary intracranial and pediatric [5,6,7,8]. In general, primary intracranial sarcomas, *DICER1*-mutant, are invariably associated with a poor prognosis. In contrast, the prognosis for extra-cranial DICER1 sarcomas with mutations in *DICER1* correlates with the histopathological grade. Here we report a new case of primary intracranial sarcomas, *DICER1*-mutant that to the best of our knowledge, is the sixth ever described in an adult.

Histopathological diagnosis of primary intracranial sarcomas, *DICER1*-mutant, is problematic because they have several morphological characteristics in common with other tumor entities. A considerable morphological overlap exits between primary intracranial sarcomas, *DICER1*-mutant, and high-grade gliomas, both tumor types being characterized by high cellularity, necrosis, vascular proliferation, a high-mitotic index, and high-grade cytological features. For this reason, the first interpretation, purely morphological, can lead to a diagnosis of a high-grade glial lesion. Even a negative immunostaining for the glial fibrillary acidic protein (GFAP) cannot exclude high-grade glioma diagnosis, as GFAP-negative glioblastomas do exist [19]. A further complication is that gliosarcoma with a diffuse mesenchymal metaplasia and without residual glial component should be considered in the differential diagnosis. The main differential diagnoses include not only gliosarcoma, but also synovial sarcoma (CD99-, CD56-positive), malignant peripheral nerve sheath tumor (usually S100- and SOX10-positive), rhabdoid tumor (desmin- and cytokeratin-positive), malignant solitary fibrous tumor (STAT6-positive, variable CD34), rabdomyosarcoma (myogenin-, muscle-specific actin-, and desmin-positive) and metastatic undifferentiated carcinomas [6,20]. To date, primary intracranial sarcomas, *DICER1*-mutant, are a molecularly defined tumor type, the diagnosis being impossible without the detection of *DICER1* mutations. Thus, immunohistochemistry should be used to exclude other primitive/metastatic malignant neoplasms a suspected mesenchymal non-meningothelial malignant tumor (i.e., sarcoma), which is best characterized through molecular analysis or methylome profiling.

Accordingly, in the case here reported, the analysis of the epigenetic profile was decisive for identifying the intracranial tumor entity: primary intracranial sarcoma, *DICER1*-mutant. Tumoral DNA sequencing confirmed the presence of *DICER1* mutations in homozygosity, allowing us to also identify the exact aminoacidic substitution. *DICER1* mutations were not detected in the patient DNA extracted from peripheral blood leucocytes and from a lymph node excised 35 years ago, excluding the case of a genetic *DICER1* syndrome.

## 4. Conclusions

In conclusion, we confirmed that DNA methylation profiling was decisive for the identification of primary intracranial sarcomas, *DICER1*-mutant, and unambiguously distinguished them from morphologically similar primary brain tumors.

## Figures and Tables

**Figure 1 brainsci-13-00235-f001:**
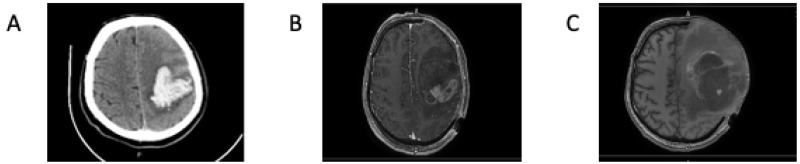
(**A**) MRI, performed in pre-surgery, showing a large, non-traumatic intraparenchymal hemorrhage with peripheral brain edema within the left temporo-parietal lobe. (**B**) MRI showing an intracranial mass after resolution of the hemorrhagic episode. (**C**) MRI showing a new hemorrhagic episode two months after surgical resection of the tumor.

**Figure 2 brainsci-13-00235-f002:**
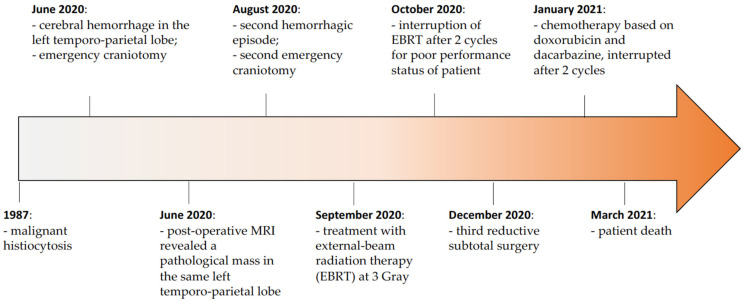
Patient timeline.

**Figure 3 brainsci-13-00235-f003:**
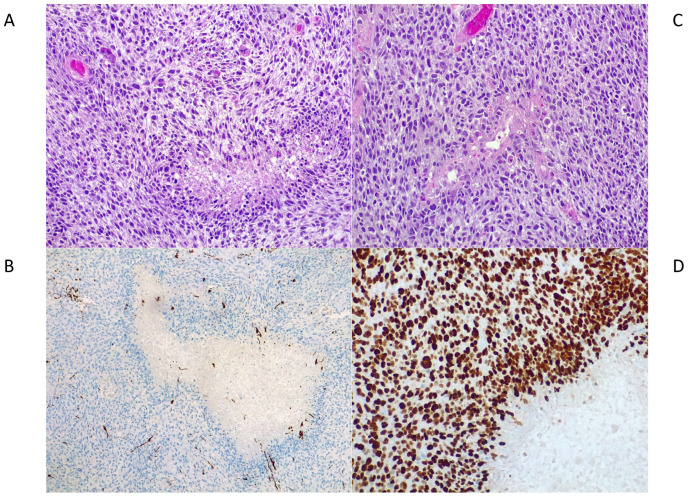
Histopathological features of the tumor. (**A**) Epithelioid to spindled cells, high mitotic activity, and a focus of palisading necrosis (hematoxylin-eosin staining, original magnification 200×). In pink the cytoplasms of cells and in violet the nuclei of cells (**B**) Vascular proliferation and perivascular pseudorosettes (hematoxylin-eosin staining, original magnification 200×) In pink the cytoplasms of cells and in violet the nuclei of cells. (**C**) SMA negativity (immunoperoxidase staining, original magnification 200×). (**D**) High proliferative index of a representative tumoral area, with Ki67 value near 70% (immunoperoxidase staining, original magnification 400×. In brown the positivity to ki67.

**Figure 4 brainsci-13-00235-f004:**
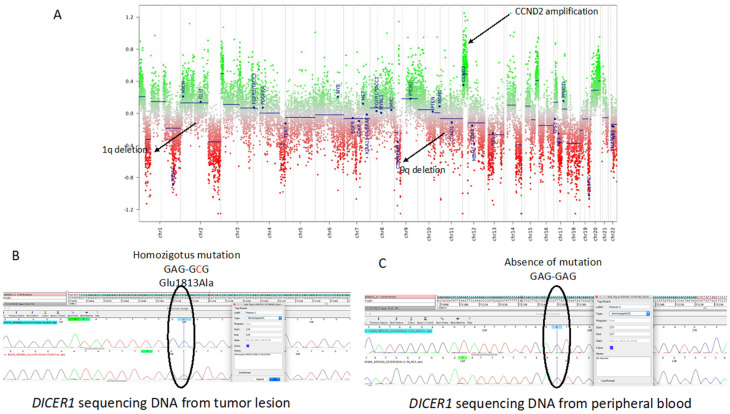
(**A**) Copy number variation (CNV) plot based on DNA methylation array data. For the calculation of CNV values, a ratio between the methylated and unmethylated data was compared with a healthy reference with a flat genome. Chromosomes are shown on the x-axis, and the log2 copy number ratio is shown on the y-axis; for every chromosome, the p-arm (**left**) and q-arm (**right**) are separated by a dotted line. Gains or amplification, in green, were calculated as positive values placed above the baseline, and deletions or losses, in red, as negative values under the baseline; in this manner, 1p and 9q deletions were clearly visible, as well as *CCND2* amplification on chromosome 12. (**B**) Sanger sequencing of the DNA of the second lesion. The electropherogram shows the homozygous mutation Glu1813Ala in exon 25 of the *DICER1* gene. (**C**) Sanger sequencing of peripheral blood DNA. The electropherogram shows wild-type *DICER1*.

**Figure 5 brainsci-13-00235-f005:**
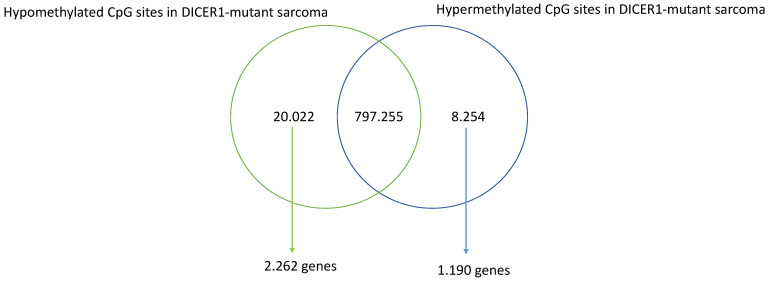
VENN diagram showing the number of CpG sites differentially methylated in primary intracranial sarcoma, *DICER1*-mutant, compared with mesenchymal glioblastoma (gliosarcoma). Raw data (IDAT files) were analyzed with RnBeads R-based scripts.

**Table 1 brainsci-13-00235-t001:** Literature review of adult primary intracranial sarcoma DICER1 mutant.

**Study**	**Age**	**Sex**	**Localization**	**Dicer Mutation**
Christian Koelsche et al. Acta Neuropathologica (2018) [6]	76 years old	female	Parieto-occipital lobe	-
Christian Koelsche et al. Acta Neuropathologica (2018) [6]	44 years old	male	Leptomeningeal supratentorial	-
Christian Koelsche et al. Acta Neuropathologica (2018) [6]	21 years old	male	Parieto-occipital lobe	-
Maki Sakaguchi et al. Brain Tumor Pathology (2019) [7]	29 years old	male	Parieto-occipital lobe	c.5127T > A, p.D1709E
Takahide Nejo et al. Clinical Medicine Insights: Case Reports (2022) [8]	69 years old	female	Frontal-lobe	c.5438A > *G* p.E1813G

## Data Availability

The data that support the findings of this study are available from the corresponding author upon reasonable request.

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
