# Peer review of "A Rare Adult Primary Intracranial Sarcoma, DICER1-Mutant Identified by Epigenomic Profiling: A Case Report"

_brainsci, 2023, doi:10.3390/brainsci13020235_

Round 1

Reviewer 1 Report

Comments and Suggestions for Authors

Dear authors, 

First of all, I’d like to give a great congratulation to them for nice and graceful study. Marinelli et al. reported a rare case of a primary brain tumor which is firstly diagnosed as a high-grade glioma using analysis of tumor DNA methylation profile. DNA sequencing confirmed epi-signature typical of spindle cell sarcomas with rhabdomyosarcoma-like features associated with mutations in DICER1 gene. They found that 48 hypomethylated genes in the DICER1 sarcoma were significantly enriched in several signal pathway involving cancer genesis such as the MAPK signaling, AMPK signaling, and cGMP-PKG signaling pathway. The topic is interesting enough to attract the reader’s interests. The analytic method is scientific and reasonable. Their reports will help lots of researchers develop new diagnostic method in classification of gliomas.

I would like to appreciate reading an interesting article. Good Luck.

Author Response

Many thanks for your positive comments. We are grateful for your opinion.

Best regards.

Reviewer 2 Report

Comments and Suggestions for Authors

1. The manuscript could benefit from editing for grammar, missing words, and subject-verb agreement, etc. It is recommended that authors delete irrelevant "general" phrases and sentences, repeated and unneeded words. They should use short sentences. Also, some Introductory sentences are irrelevant or are not needed. It is also recommended that authors send their manuscript to an expert in English editing and academic writing. For example, this sentence is incomplete and needs revision and rephrasing “Primary malignant mesenchymal brain tumors are a diagnostic challenge for pathologists, because rare and heterogeneous.”

2. Title: DICER1 should be italicized. In scientific writing, in general, symbols for genes are italicized whereas symbols for proteins are not italicized. The formatting of symbols for RNA and complementary DNA (cDNA) usually follows the same conventions as those for gene symbols. Gene names that are written out in full are not italicized (e.g., insulin-like growth factor 1). Genotype designations should be italicized, whereas phenotype designations should not be italicized. Please review the manuscript and italicize all gene names.

3. All abbreviations should be revised and defined at their first use. For example, NOS in the abstract should be defined.

4. When reporting a case, authors should follow the CAse REports (CARE) or Surgical CAse REports (SCARE) guidelines. In this case reports, authors did not follow all of those guidelines.

5. Keywords: According to the CARE/SCARE guidelines, 3 to 6 key words that identify areas covered in the case report (including "case report" as one of the keywords) should be used.

6. Abstract: The abstract should be a total of about 200 words maximum. Please reduce the word count to 200 instead of 267.

7. Abstract: “Here we report a case of a primary” please add a comma after Here.

8. Abstract: “typical of spindle cell sarcomas with” change sarcomas to sarcoma.

9. Abstract: “DICER1 gene (SCS-RMSlike-DICER1).” Italicize DICER1 and correct the gene name: SCS-RMS like-DICER1.

10. Abstract: “information about this so rare tumor” change “so rare” to “extremely rare.”

11. Abstract: “characterization of the fifth so far described case.” If only three other cases have been reported in literature, why did the authors say, “fifth so far described case”? In fact, there are more than 80 cases of DICER-1 mutant described in literature (https://pubmed.ncbi.nlm.nih.gov/31537896/; https://pubmed.ncbi.nlm.nih.gov/32291395/; https://pubmed.ncbi.nlm.nih.gov/31487013/; https://pubmed.ncbi.nlm.nih.gov/30649606/; https://pubmed.ncbi.nlm.nih.gov/29881993/).

12. Abstract: “Taken together, we here” remove ‘here’.

13. Abstract: According to the CARE guidelines, the abstract should answer the following: what is unique or educational about the case? what does it add to the medical literature? why is this important? What are the patient's main concerns and important clinical findings? What is the main diagnosis, therapeutics interventions, and outcomes? And what are the “take-away” lessons from this case? Some of those questions are not answered in this case.

14. Introduction: The introduction should be restructured into several paragraphs instead of one paragraph. In fact, according to CARE/SCARE guidelines, it is better to have 2 paragraphs.

15. Introduction: According to the CARE guidelines, the introduction should include a summary of why this case is unique or educational with reference to the relevant medical literature and current standard of care (with references, 1-2 paragraphs). Besides, nature of the institution in which the patient was managed should be indicated: academic, community or private practice setting?

16. The manuscript lacks a timeline. According to the CARE guidelines, a timeline should be designed including data which allows readers to establish the sequence and order of events in the patient's history and presentation (using a table or figure if this helps). Delay from presentation to intervention should also be reported.

17. According to the latest WHO classification of CNS tumors, the correct name of this entity is “primary intracranial sarcoma, DICER1-mutant.” Consider correcting it throughout the manuscript. Also, more importantly, it is not recommended to use this terminology: primary intracranial spindle cell sarcoma with rhabdomyosarcoma-like features, DICER1-mutant.

18. The term “SCS-RMSlike-DICER1” should not be used throughout the manuscript as it is not recommended by the WHO latest classification of CNS tumors. Also, authors need to cite the latest WHO classification.

19. Results: It would be interesting to add the imaging studies performed including the MRI and CT findings.

20. Results: Figure 1 is not very clear. Consider using higher resolution images. Also, the letters a, b, c, and d should be capitalized and located on the upper left corner of the images instead of lower right.

21. Results: The legend of the figure is not very well described. Consider rephrasing. For example, “perivascular arrangement of neoplastic cells” could be further described as “pseudorosette” which is perivascular radial arrangement of neoplastic cells around a small blood vessel. The Ki67 proliferation index appears higher than 70% (almost 90-99%).

22. Results: Figure 2 is also not clear. Specifically, parts b and c are not clear at all. In fact, it is not clear what the authors want to show in this figure.

23. According to the CARE guidelines, patient info, clinical findings, diagnostic assessment, therapeutic intervention, and follow-up and outcome should be complete. In this case, some of that info are missing.

24. Authors should have a scientific discussion of the strengths AND limitations associated with this case report, relevant medical literature with references, and scientific rationale for any conclusions (including assessment of possible causes).

25. A table summarizing other cases reported in literature can be added.

Author Response

Dear Reviewer #2,

please consider this thoroughly revised version of our manuscript entitled “A rare adult primary intracranial sarcoma, DICER1-mutant identified by epigenomic profiling: a case report”, number brainsci-2143702, by Marinelli et al. Please note that, as suggested by you and to follow the CARE/SCARE guidelines, the manuscript title has been changed from the original “A rare case of intracranial adult DICER1 mutant sarcoma individuated 
by epigenomic profiling”.

We wish to thank you for the favourable evaluation of our manuscript and, mostly, for your punctual criticisms that significantly helped to improve it.

A point-to-point reply to your comments follows. For your convenience, all the changes have been tracked in the “Marinelli et al. revised” file.

Thank you again for the attention you have dedicated to our work.

Sincerely,

Rosa Della Monica, on behalf of all authors.

Reviewer 2

  1. The manuscript could benefit from editing for grammar, missing words, and subject-verb agreement, etc. It is recommended that authors delete irrelevant "general" phrases and sentences, repeated and unneeded words. They should use short sentences. Also, some Introductory sentences are irrelevant or are not needed. It is also recommended that authors send their manuscript to an expert in English editing and academic writing. For example, this sentence is incomplete and needs revision and rephrasing “Primary malignant mesenchymal brain tumors are a diagnostic challenge for pathologists, because rare and heterogeneous.”

As suggested, we have extensively revised the manuscript text. Finally, the new version of the manuscript has been edited by the MDPI English editing service.

  1. Title: DICER1 should be italicized. In scientific writing, in general, symbols for genes are italicized whereas symbols for proteins are not italicized. The formatting of symbols for RNA and complementary DNA (cDNA) usually follows the same conventions as those for gene symbols. Gene names that are written out in full are not italicized (e.g., insulin-like growth factor 1). Genotype designations should be italicized, whereas phenotype designations should not be italicized. Please review the manuscript and italicize all gene names.

As suggested, we have italicized all gene names throughout the manuscript.

  1. All abbreviations should be revised and defined at their first use. For example, NOS in the abstract should be defined.

As suggested, we have defined all the abbreviations at their first use.

  1. When reporting a case, authors should follow the CAse REports (CARE) or Surgical CAse REports (SCARE) guidelines. In this case reports, authors did not follow all of those guidelines.

As suggested, we have reorganized the manuscript according to CARE/SCARE guidelines. We have: 1) added the word “case report” in the title; 2) revised the keywords and included the word “case report”; 3) changed the Results section into a paragraph entitled “Case Presentation” (please, see page 2 lane 58); 4) removed the Materials and Method section summarizing the used protocols in the Case Presentation section.   

  1. Keywords: According to the CARE/SCARE guidelines, 3 to 6 key words that identify areas covered in the case report (including "case report" as one of the keywords) should be used.

As suggested, we have revised the keywords and included the word “case report”.

  1. Abstract: The abstract should be a total of about 200 words maximum. Please reduce the word count to 200 instead of 267. 

As suggested, we have abbreviated the abstract (from the original 267 to 182 words).

  1. Abstract: “Here we report a case of a primary” please add a comma after Here.

As suggested, we have added the comma (please, see page 2, line 49).

  1. Abstract: “typical of spindle cell sarcomas with” change sarcomas to sarcoma.

Done as suggested (please, see page 2, line 51).

  1. Abstract: “DICER1 gene (SCS-RMSlike-DICER1).” Italicize DICER1 and correct the gene name: SCS-RMS like-DICER1.

Done as suggested.

  1. Abstract: “information about this so rare tumor” change “so rare” to “extremely rare.”

Done as suggested.

  1. Abstract: “characterization of the fifth so far described case.” If only three other cases have been reported in literature, why did the authors say, “fifth so far described case”? In fact, there are more than 80 cases of DICER-1 mutant described in literature (https://pubmed.ncbi.nlm.nih.gov/31537896/; https://pubmed.ncbi.nlm.nih.gov/32291395/; https://pubmed.ncbi.nlm.nih.gov/31487013/; https://pubmed.ncbi.nlm.nih.gov/30649606/; https://pubmed.ncbi.nlm.nih.gov/29881993/).

We apologize for not having been clear, evidently. By saying the “fifth so far described case”, we intended to mention only the intracranial DICER1-mutant sarcomas found in adult patients. So far, the majority of DICER1-mutant sarcomas described in the literature have been diagnosed in children and some of that were not intracranial tumors. Please also note that, during the manuscript revision process, another adult case of DICER1-mutant sarcoma has been described (reference #8). Thus, we have made sure that, in the revised manuscript, is now clear that our case is the sixth intracranial DICER1-mutant sarcoma found in adults (please, see page 2, lines 52-54).

  1. Abstract: “Taken together, we here” remove ‘here’.

Done as suggested.

  1. Abstract: According to the CARE guidelines, the abstract should answer the following: what is unique or educational about the case? what does it add to the medical literature? why is this important? What are the patient's main concerns and important clinical findings? What is the main diagnosis, therapeutics interventions, and outcomes? And what are the “take-away” lessons from this case? Some of those questions are not answered in this case.

We have revised the Abstract according to your suggestions.

  1. Introduction: The introduction should be restructured into several paragraphs instead of one paragraph. In fact, according to CARE/SCARE guidelines, it is better to have 2 paragraphs.

Done as suggested. See also below (point 15).

  1. Introduction: According to the CARE guidelines, the introduction should include a summary of why this case is unique or educational with reference to the relevant medical literature and current standard of care (with references, 1-2 paragraphs). Besides, nature of the institution in which the patient was managed should be indicated: academic, community or private practice setting?

As suggested, we have divided the Introduction into two paragraphs, separated by a full stop. We have also highlighted why this case is educational. Moreover, we have added the relevant references. Finally, we have specified that the patient was managed in an academic institution.

  1. The manuscript lacks a timeline. According to the CARE guidelines, a timeline should be designed including data which allows readers to establish the sequence and order of events in the patient's history and presentation (using a table or figure if this helps). Delay from presentation to intervention should also be reported.

As suggested, in the revised version of the manuscript, we have added a new figure (new Figure 2) describing the history of the patient.

  1. According to the latest WHO classification of CNS tumors, the correct name of this entity is “primary intracranial sarcoma, DICER1-mutant.” Consider correcting it throughout the manuscript. Also, more importantly, it is not recommended to use this terminology: primary intracranial spindle cell sarcoma with rhabdomyosarcoma-like features, DICER1-mutant.

As suggested, and according to the latest WHO classification, we have corrected the tumor name throughout the manuscript, using “primary intracranial sarcoma, DICER1-mutant”. Also, we have deleted throughout the manuscript the terminology “primary intracranial spindle cell sarcoma with rhabdomyosarcoma-like features, DICER1-mutant”.

  1. The term “SCS-RMSlike-DICER1” should not be used throughout the manuscript as it is not recommended by the WHO latest classification of CNS tumors. Also, authors need to cite the latest WHO classification.

As suggested, and according to the latest WHO classification, we have corrected the tumor name throughout the manuscript, using “primary intracranial sarcoma, DICER1-mutant”. Also, we have deleted throughout the manuscript the term “SCS-RMSlike-DICER1”. Finally, we have cited the latest WHO classification (reference #11).

  1. Results: It would be interesting to add the imaging studies performed including the MRI and CT findings.

As suggested, we have added a new figure (new Figure 1) showing the MRIs of the patient.

  1. Results: Figure 1 is not very clear. Consider using higher resolution images. Also, the letters a, b, c, and d should be capitalized and located on the upper left corner of the images instead of lower right.

Done as suggested.

  1. Results: The legend of the figure is not very well described. Consider rephrasing. For example, “perivascular arrangement of neoplastic cells” could be further described as “pseudorosette” which is perivascular radial arrangement of neoplastic cells around a small blood vessel. The Ki67 proliferation index appears higher than 70% (almost 90-99%).

As suggested, we have rephrased the legend of the figure showing the morphological feature of the tumor (new Figure 3). The figure showing the Ki67 proliferation index is a representative region of the tumor in which, as you rightly noted, about 90% of the cells were positive to Ki67. However, the Ki67 index has been calculated as the mean of the entire tumoral area, resulting in 70% of positivity.  

  1. Results: Figure 2 is also not clear. Specifically, parts b and c are not clear at all. In fact, it is not clear what the authors want to show in this figure.

Done as suggested. We have now indicated in the Figure CNVs alterations and hot spot mutations.

  1. According to the CARE guidelines, patient info, clinical findings, diagnostic assessment, therapeutic intervention, and follow-up and outcome should be complete. In this case, some of that info are missing.

We have added a schematic timeline of the patient (new Figure 2) and added a new paragraph (#2.2) describing all the patient information.

  1. Authors should have a scientific discussion of the strengths AND limitations associated with this case report, relevant medical literature with references, and scientific rationale for any conclusions (including assessment of possible causes).

We have carefully revised the discussion section, as the rest of the manuscript. We believe that the discussion now meets all the above specified requirements.

  1. A table summarizing other cases reported in literature can be added.

As suggested, we have added a table summarizing the cases reported in the literature (new Table 1; page 2, line 54).

Round 2

Reviewer 2 Report

Comments and Suggestions for Authors

Figure 4 is not present in the manuscript. Please add it. Also, I am wondering why is the font different in the abstract and Table 1.

One important thing is that the authors did not make changes as tracked changes or highlighted to track them. This makes it impossible for a reviewer to track the changes. Please resubmit using tracked changes so that reviewers can know what changes were made.